# COVID-19 stigmatization after the development of effective vaccines: Vaccination behavior, attitudes, and news sources

**Don C. Des Jarlais**[1,2]*, **Sarah Lieff**[2], **Margaux Grivel**[2], **Gabriella Meltzer**[2], **Jasmin Choi**[2], **Chenziheng Allen Weng**[3], **Jonathan P. Feelemyer**[4], **Virginia W. Chang**[2,4‡], **Lawrence Yang**[2,5‡]

**1** Department of Epidemiology, New York University School of Global Public Health, New York, NY, United States of America, **2** Department of Social & Behavioral Sciences, New York University School of Global Public Health, New York, NY, United States of America, **3** Department of Biostatistics, New York University School of Global Public Health, New York, NY, United States of America, **4** Department of Population Health, New York University Grossman School of Medicine, New York, NY, United States of America, **5** Department of Epidemiology, Columbia University Mailman School of Public Health, New York, NY, United States of America

‡ VWC and LY are co-senior authorship to this work.
* Don.DesJarlais@nyu.edu

**Data Availability Statement:** There are important ethical and legal restrictions to publicly sharing the data from this manuscript, and we will note this in the updated data availability statement. Our survey

## Abstract

### Objective

To compare COVID-19 stigmatization at two pandemic time points (1) August 2020—during lockdowns and prior to vaccine rollout, and (2) May 2021—during vaccine rollout, when approximately half of U.S. adults were vaccinated.

### Methods

Comparison of COVID19-related stigmatization and associated factors in two national internet surveys conducted in August 2020 (N = 517) and May 2021 (N = 812). Factors associated with endorsing stigmatization were identified using regression analysis. The main outcomes included endorsement of stigmatization and behavioral restrictions towards persons with COVID-19 and towards persons of Chinese descent. A previously developed "stigmatizing attitudes and behavioral restrictions" scale was adapted to measure the intersection of negative attitudes toward COVID-19 disease and negative attitudes toward persons of Chinese descent.

### Results

COVID-19 related stigmatization declined significantly from August 2020 to May 2021. Many factors were associated with stigmatizing in both surveys: full time employment, Black race, Hispanic ethnicity, worry about contracting COVID-19, probable depression, and Fox News and social media as sources of information (all positively associated), and self-assessed knowledge about COVID-19, contact with Chinese individuals, and publicly

contains Personal Health Information (PHI) which is protected under the Health Insurance Portability and Accountability Act (HIPPA), such as vaccination status and whether the individual suffers from any of the "underlying conditions" that would be likely to make a COVID infection more severe, as well as demographic characteristics. Providing public access to the data used in our analyses could threaten loss of the confidentiality of PHI, and was not to be permitted according to our proposal submitted to the IRB. Access to the data can be provided through an approved Data Use Agreement between our institution (New York University) and the institution with which the user is affiliated. Persons wanting to access the data should communicate with the NYU IRB (email contact: irbinfo@nyulangone.org) to initiate a Data Use Agreement.

**Funding:** This work was supported by New York University and was funded through a research grant from New York University's Anti-Racism Center (20-61518-G1336-MC1039).

**Competing interests:** The authors have declared that no competing interests exist.

funded news as sources (all negatively associated). Positive attitudes toward vaccination were associated with stigmatization.

## Conclusions

COVID-19 related stigmatization reduced substantially over these two points in the pandemic, with many continuities in the factors associated with stigmatizing. Despite the reduction in stigmatizing, however, some stigmatizing attitudes for both COVID-19 and Chinese individuals remained.

## Introduction

Control of infectious diseases often requires identifying persons who have been exposed to or contracted the disease and then placing restrictions on their behavior. Isolation of active cases and quarantine of persons who have been exposed are two examples of such identification and behavioral restrictions. While such identification and restrictions may be critical for controlling the disease, they may also lead to stigmatization of persons with the disease and social groups associated with the disease.

A critical distinction between good public health practice and stigmatization is that in good public health practice, the persons undergoing behavioral restrictions are treated with dignity, respect, and support, while in stigmatization, persons are often treated with hostility and contempt. A second critical distinction is the public health behavioral restrictions are evidence-based, while stigmatization is often based in exaggerated fears and pre-existing negative attitudes towards specific social groups. Stigmatization can cause substantial psychological, social, physical, and economic harm to individuals and groups that are stigmatized [1–3].

Stigmatization may also lead to avoiding medical care, attempting to hide disease and exposures, and to hiding membership in groups associated with the disease, all of which may contribute to further transmission [4]. The negative consequences of infectious disease related stigmatization may be particularly likely if the stigmatization reinforces existing negative social stereotypes of the groups associated with the disease [5, 6].

Early in the SARS-CoV-2/COVID-19 epidemic, there was considerable stigmatization of persons with COVID-19 and persons of Chinese descent (and in particular persons from Wuhan) [7]. Some political leaders in the United States (U.S.) linked the SARS-CoV-2 virus to China through the use of derogatory terms such as "China virus," "Wuhan virus," and "Kung flu" [8]. This terminology clearly linked the disease to persons of Chinese nationality and the linkage occurred in parallel with an over 150% increase in anti-Asian hate crimes reported in the U.S. in 2020–2021 compared to the previous year and reports of Asians experiencing direct and indirect racial hostility [9–11]. Other countries have also reported anti-Chinese/anti-Asian sentiment associated with COVID-19; examples including Australia [12], Europe [13] and South Korea [14].

Highly effective vaccines that greatly reduce serious illness and deaths from COVID-19 have been developed and made widely available in the U.S. Approximately 62% of adults in the U.S. had received at least one dose of a vaccine by the time of our second data collection (late May 2021) and 77.6% had received at least one dose as of May 3, 2022, a year later [15].

The development and partial implementation of the vaccines, however, has been followed by intense controversy [16–19]. The controversy has often been fed by the considerable

amount of misinformation and disinformation about COVID-19 that has been disseminated in the U.S., often through social media platforms [20–23].

In this study, we used non-random national internet samples to compare the COVID-19 stigmatization U.S. at two different points in the epidemic. The first survey was conducted in August 2020, when various lockdowns had been implemented [24]. The second survey occurred in May 2021, a time when approximately half of U.S. adults had received at least one dose of a COVID-19 vaccine, and the Centers for Disease Control and Prevention (CDC) had just announced guidance that fully vaccinated persons no longer needed to wear masks in most settings [25]. This time period also coincided with increasing public polarization over the vaccines [26, 27]. We consider two primary questions in the comparison of the two surveys: 1) Did the overall frequency of stigmatization change? and 2) Did the factors associated with expressing stigmatization remain the same or change? Identifying changes and consistencies in stigmatization within a rapidly changing pandemic may provide insights into the nature of disease-related stigmatization and possibly for preparing for future epidemics and pandemics. The Health Belief Model (HBM) is used to interpret the findings [28, 29]. This model is frequently used in health communications, in which the perceived threat (seriousness and susceptibility) of the disease is emphasized as a rationale for engaging in protective behaviors. The "protective behaviors" may include avoiding contact with and placing behavioral restrictions on persons with the disease. However, given the controversies around the vaccines and the mis- and dis-information about the vaccines, we did not attempt to use the HBM to predict how stigmatization and how the correlates of stigmatization might have changed from pre- to post development of the vaccines.

## Methods

### Data collection

Amazon's Mechanical Turk (MTurk) is a crowdsourcing platform that provides access to an on-demand and diverse global workforce, an estimated 225,000 of which are U.S.-based [30]. Through MTurk, researchers and other "requesters" can post tasks that workers can complete if they meet pre-specified criteria. Participants in this study were eligible to complete the online survey if they were at least 18 years of age, had previously completed at least 500 MTurk tasks, and had approval ratings greater than or equal to 90% for previously completed MTurk tasks. The first survey was launched on August 5, 2020, during a period of intense lockdowns and social distancing. The second survey was launched on May 21, 2021, soon after the CDC announced new guidance that fully vaccinated persons no longer needed to wear masks or practice social distancing outdoors or in most indoor settings, excluding health care facilities, while flying or taking public transit, and in congregate settings [25]. Persons who had not been vaccinated, however, were still advised to wear masks in almost all situations outside of private homes. The May 2021 survey included all the items in the August 2020 survey, and additional items about having been vaccinated and intentions to become vaccinated.

Data quality was assessed using two attention checks requiring participants to select a specific response option and a time-to-completion analysis. Given the significant outliers, we implemented a minimum acceptable time to complete the survey of one standard deviation below the whole sample mean (151.562 seconds for the August 2020 survey, 182.897 seconds for May 2021 survey), and a maximum cutoff score of 1,000 seconds for the August 2020 survey and 1,400 seconds for the May 2021 survey. Of the 1,606 respondents between the two surveys, a final sample of 1,312 (August 2020 N = 517; May 2021 N = 812) was retained for this analysis. Entries were excluded due to failed attention checks (N = 51), unacceptable completion times (N = 64), and having incomplete data (N = 179).

## Study variables

As a primary purpose of the 2021 survey was to compare COVID-19 related stigmatization at two points in the pandemic—during initial lockdown vs. after vaccine rollout—we used many of the items from the 2020 survey in the 2021 survey. Table 1 presents the major COVID-19 related items in our two surveys.

## Primary outcome

The primary outcome of interest was COVID-19 related stigmatization, measured with the Stigmatizing Attitudes and Behavioral Restrictions (SABR) Scale. This SABR scale was adapted from a 2003 study comparing stigmatization of SARS-1 and HIV/AIDS [31]. Despite the many differences in the epidemiology of SARS-1 and HIV/AIDS (in modes of transmission, in case fatality rates, geographic areas of concentration, social groups with which the diseases were associated, total numbers of cases) there were strong similarities in the patterns of responses to the SABR items for SARS-1 and HIV/AIDS.

Responses to all of the individual SABR items were significantly correlated across the two diseases, and many of the same factors were associated with stigmatizing both SARS-1 and HIV/AIDS. These associated factors included education, income, race/ethnicity, greater worry about contracting the disease, less knowledge of the disease, and mental health problems [31].

Exploratory factor analysis of the August 2020 survey showed a strong single general factor explaining approximately 71% of the variance, with all items loading 0.6 or greater onto this factor. Reliability for this scale was excellent (Cronbach's alpha = 0.90). This SABR scale should be considered as measuring the intersection of stigmatization of COVID-19 and stigmatization of persons of Chinese descent. It was not intended to capture all aspects of stigmatization of either the disease or of the social group.

## Contact with persons with or associated with COVID-19 and with Chinese persons

These items were adapted from stigma research [32] suggesting that personal contact would decrease stigmatization.

## Experiences of discrimination related to COVID-19 and race/ethnicity

We included questions on having experienced stigmatization to examine whether experiencing stigmatization might lead to empathy for persons associated with COVID-19 and less stigmatizing or might lead to resentment and increased stigmatizing.

## Other COVID-19 related questions

Several additional questions were adapted from Health Belief Model [28, 29] theory (which posits that a perceived threat of a disease will lead to greater motivation to actions that would reduce the threat of the disease) and the 2003 study that suggested the threat value of the disease (worry about contracting the disease) will be associated with efforts to avoid contact (placing behavioral restrictions on persons with or associated with the disease). Probable depression was also considered likely to increase the threat value of the disease. Knowledge about the disease is clearly an important aspect of beliefs about the disease, and our 2020 survey found that self-assessed knowledge was associated with reduced stigmatization. New items were added on having underlying conditions that would increase the likelihood of severe COVID-19 illness and contact with persons who had severe COVID-19 as we hypothesized these would increase the perceived severity of the COVID-19.

**Table 1. List of different COVID-19 related questions and variables and recoding schemes for analysis.**

| | Original Scale Scores | Recoded Variable |
|---|---|---|
| **Stigmatizing Attitudes and Behavioral Restrictions (SABR) Scale items** | | |
| 1. Requiring Americans with COVID-19 to wear identification tags | 1. Agree Strongly 2. Agree Somewhat 3. Disagree Somewhat 4. Disagree Strongly | 1. "No Stigma" if answered "disagree" to all SABR items 2. "Any Stigma" if answered agree to any SABR items |
| 2. The government announcing it will execute people who knowingly spread COVID-19 | | |
| 3. Avoiding areas in the United States that are heavily populated by Chinese individuals | | |
| 4. Forcing all Chinese people to be medically checked for COVID-19 | | |
| 5. Not allowing Chinese people to enter the United States | | |
| **Contact with Persons Questions** | | |
| Have you ever known somebody who had COVID-19 | 1. Definitely Yes 2. Probably Yes 3. Probably No 4. Definitely No | 1. Contact (1,2) 2. No Contact (3,4) |
| Have you ever known somebody who became seriously ill or died from COVID-19 | | |
| Have you ever known somebody who identifies as Chinese? | | |
| **Experience of Discrimination related to COVID-19** | | |
| Have you experienced stigmatization or discrimination related to COVID-19? | 1. Yes, a lot 2. Some 3. No | 1. Yes, a lot/Some (1,2) 2. No (3) |
| Have you experienced stigmatization or discrimination because of your race/ethnicity | | |
| **Vaccination Behavior and Intentions** | | |
| Have you been or are you in the process of being vaccinated against COVID-19, and if not, do you intend to receive a COVID-19 vaccination in the future | 1. Vaccinated 2. Definitely will get vaccinated 3. Probably will get vaccinated 4. Probably will not get vaccinated 5. Definitely will not get vaccinated | 1. Vaccinated (1) 2. Definitely/probably will get vaccinated (2,3) 3. Probably will not get vaccinated (4) 4. Definitely will not get vaccinated (5) Further combined as 1.Pro-vaccines (1, 2, 3) and 2. Anti-vaccines (4, 5) |
| **Other COVID-19 related questionnaire items** | | |
| Do you have any of the underlying conditions, such as diabetes, being overweight, heart disease, lung/breathing diseases, that could make COVID-19 disease more severe | 1. Yes 2. No | 1. Yes if answered for any condition 2. No if answered no for all conditions |
| Probable current depression | 1. PHQ score | 1. Yes if PHQ score of 5 or more 2 No if PHQ score of less than 5 |
| How worried are you about contracting COVID-19? | 3. Not at all worried 4. Somewhat worried 5. Very worried | 1. Not at all worried (1) 2. Somewhat/very worried (2,3) |
| How much have you heard about COVID-19? (Self-assessed knowledge) | 1. Not much 2. Some 3. A great deal | 1. Not much/Some knowledge (1,2) 2. A great deal of knowledge (3) |

(*Continued*)

**Table 1.** (Continued)

| | Original Scale Scores | Recoded Variable |
|---|---|---|
| News Sources | 1. Facebook<br>2. Twitter<br>3. NPR<br>4. PBS<br>5. CNN<br>6. MSNBC<br>7. ABC<br>8. CBS<br>9. NBC<br>10. Other News source<br>11. New York Times<br>12. Fox News | 1. Social Media (1,2)<br>2. Publicly-funded News (These are stations funded through government sources and donations) (3,4)<br>3. Commercial TV News (These are stations funded through advertising) (5–9)<br>4. Other news source (10) |

New York Times (11) and Fox News (12) were retained as separate news variables because of prior research indicating their importance in the coverage of COVID-19 news [33–35]

The vaccination behavior and intentions items were added to the May 2021 survey to examine the relationship of being vaccinated/not being vaccinated to stigmatizing.

## Preferred news source

These items were used in the 2020 survey and were strongly associated with stigmatization in that survey. That preferred news sources are associated with beliefs about COVID-19 has been noted in many previous studies [36–38].

## Lack of hypotheses

We did not formulate hypotheses for our two major research questions (whether the prevalence of stigmatization or the correlates of stigmatization would change after development of vaccines) because we believed that the amount of polarization, mis- and dis-information about the vaccines precluded straightforward application of any theoretical framework.

## Statistical analysis

Cross tabulations and chi square tests assessed bivariate relationships between the August 2020 and May 2021 surveys and demographic characteristics, vulnerabilities to and experiences with COVID-19, and the SABR scale, with p-values of less than 0.05 used to detect statistically significant differences. We calculated crude odds ratios to examine associations of possible predictor variables with the COVID-19 SABR items using univariate logistic regression. All predictors and covariates were moved into multivariable logistic regression models to estimate adjusted odds ratios. The vaccination status/vaccination intentions variable formed a continuum from positive to very negative attitudes toward vaccination, and the Cochran-Armitage test for trend was used to assess relationships with COVID-19 related attitudes and experiences and the SABR items.

As noted below, there were modest differences in the demographic characteristics of the respondents in the two surveys. We thus used demographic characteristics as control variables in all multivariable analyses.

Analyses were performed in STATA version 17 [39]. The study was approved by the New York University Institutional Review board. All participants provided written informed consent after reading a summary of the study, but before beginning the survey.

## Results

Table 2 presents the demographic characteristics of respondents in the two surveys. There were modest differences in the demographic characteristics of the participants in the two surveys. Participants in the May 2021 survey were moderately older, less likely to be non-Hispanic Black or Hispanic, and less likely to have college degrees.

Table 3 presents endorsements of the five SABR items in the two surveys. In the May 2021 survey, there was less endorsement of the SABR items, including for the scale as a whole and for each of the individual items. All the differences in the SABRs were in the direction of less stigmatization in the May 2021 survey.

The changes in stigmatization occurred within the context of changes in many other aspects of the COVID-19 pandemic. Table 4 presents information on experiences with and potential vulnerabilities to COVID-19 among the respondents. There were multiple differences in the responses to these survey items. Participants in the May 2021 survey were less likely to be worried about contracting COVID-19, more likely to assess themselves as knowledgeable, more likely to report contact with persons who had COVID-19, less likely to report probable depression, and less likely to report social media as a source of information.

**Table 2. Sample characteristics for Waves 1 (N = 517) and 2 (N = 812).**

| | Wave 1 | | Wave 2 | | p-value |
|---|---|---|---|---|---|
| | N = 517 | | N = 812 | | |
| | N | % | N | % | |
| Gender | | | | | |
| Female | 186 | 36.0 | 324 | 39.9 | 0.15 |
| Male | 331 | 64.0 | 488 | 60.1 | |
| Age | | | | | |
| 18–24 years | 26 | 5.0 | 26 | 3.2 | 0.04 |
| 25–34 years | 242 | 46.8 | 332 | 40.9 | |
| 35–44 years | 149 | 28.8 | 253 | 31.2 | |
| 45–54 years | 55 | 10.6 | 110 | 13.6 | |
| 55+ | 45 | 8.7 | 91 | 11.2 | |
| Race | | | | | |
| NH White | 295 | 57.1 | 542 | 66.8 | <0.001 |
| NH Black | 69 | 13.4 | 65 | 8.0 | |
| Hispanic | 137 | 26.5 | 151 | 18.6 | |
| Asian | 16 | 3.1 | 54 | 6.7 | |
| Employment | | | | | |
| No employment | 44 | 8.5 | 89 | 11.0 | 0.33 |
| Part-time | 61 | 11.8 | 89 | 11.0 | |
| Full-time | 412 | 79.7 | 634 | 78.1 | |
| Education | | | | | |
| High School or Below | 55 | 10.6 | 93 | 11.5 | 0.01 |
| Some College | 71 | 13.7 | 161 | 19.8 | |
| College Degree (BA/BS) | 317 | 61.3 | 433 | 53.3 | |
| Graduate School | 74 | 14.3 | 125 | 15.4 | |

**Table 3. Endorsements of the five SABR items for Waves 1 (N = 517) and 2 (N = 812).**

| | Wave 1 | | Wave 2 | | p-value |
|---|---|---|---|---|---|
| | N = 517 | | N = 812 | | |
| Should people avoid areas in the United States heavily populated by Chinese? | | | | | |
| Agree/Strongly agree | 248 | 48.0 | 269 | 33.1 | <0.001 |
| Disagree/Strongly disagree | 269 | 52.0 | 543 | 66.9 | |
| Should all Chinese by forcibly checked for COVID-19? | | | | | |
| Agree/Strongly agree | 235 | 45.5 | 242 | 29.8 | <0.001 |
| Disagree/Strongly disagree | 282 | 54.6 | 570 | 70.2 | |
| Should Chinese not be allowed to enter the United States? | | | | | |
| Agree/Strongly agree | 261 | 50.5 | 253 | 31.2 | <0.001 |
| Disagree/Strongly disagree | 256 | 49.5 | 559 | 68.8 | |
| Should Americans with COVID-19 be required to wear identification tags? | | | | | |
| Agree/Strongly agree | 257 | 49.7 | 288 | 35.5 | <0.001 |
| Disagree/Strongly disagree | 260 | 50.3 | 524 | 64.5 | |
| Should people who knowingly spread COVID-19 be executed? | | | | | |
| Agree/Strongly agree | 235 | 45.5 | 262 | 32.3 | <0.001 |
| Disagree/Strongly disagree | 282 | 54.6 | 550 | 67.7 | |
| Have you experienced stigmatization or discrimination because of your race/ethnicity? | | | | | |
| Yes, a lot/some | | | 298 | 36.7 | |
| No | | | 514 | 63.3 | |
| COVID Stigma (5-item composite)[1] | | | | | |
| Some stigma | 341 | 66.0 | 377 | 46.4 | <0.001 |
| No stigma | 176 | 34.0 | 435 | 53.6 | |

[1] COVID Stigma 5-item composite includes the following items

Should Americans with COVID-19 be required to wear identification tags?

Should people who knowingly spread COVID-19 be executed?

Should people avoid areas in the United States heavily populated by Chinese?

Should all Chinese be forcibly checked for COVID-19?

Should Chinese not be allowed to enter the United States?

We then conducted a bivariable logistic regression of the odds of stigmatizing in the May 2021 survey compared to the odds of stigmatizing in the August 2020 survey. The OR was 0.45 (95% CI 0.36–0.56). We also conducted a multivariable logistic regression to control for the demographic and "vulnerabilities and experiences" variables in our survey; the adjusted OR was 0.52 (95% CI 0.37–0.72) showing that the reduction in endorsing SABR items remained significant after controlling for the other variables in our survey. The odds for endorsing at least one SABR item were 48% lower in the May 2021 survey compared to the August 2020 survey. (Full data available from the first author).

## Factors associated with endorsing SABRs in August 2020 and May 2021 surveys

We used logistic regression to identify factors associated with the SABR scale in the two surveys. Table 5 presents the bivariable and multivariable models for each of the two surveys. Overall, there were many variables that were significant in at least one of the regressions and substantial similarities in the factors that were significantly associated with the SABR scale across the two surveys. The variables significantly associated with stigmatization in both surveys were: full time employment, Black race, Hispanic ethnicity, worry about contracting

**Table 4. COVID-19 related experiences and vulnerabilities for Waves 1 (N = 517) and 2 (N = 812).**

| | Wave 1 | | Wave 2 | | p-value |
|---|---|---|---|---|---|
| | N = 517 | | N = 812 | | |
| | **N** | **%** | **N** | **%** | |
| COVID Knowledge | | | | | |
| Some | 89 | 17.2 | 107 | 13.2 | 0.04 |
| A great deal | 428 | 82.8 | 705 | 86.8 | |
| COVID Worry | | | | | |
| Somewhat/very worried | 459 | 88.8 | 620 | 76.4 | <0.001 |
| Not at all worried | 58 | 11.2 | 192 | 23.7 | |
| Do you have any of the underlying conditions, such as diabetes, overweight, heart disease, lung/breathing diseases, that could make COVID-19 disease more severe? | | | | | |
| Yes | | | 309 | 38.0 | |
| No | | | 503 | 62.0 | |
| Contact with Chinese | | | | | |
| Contact | 390 | 75.4 | 648 | 79.8 | 0.06 |
| No contact | 127 | 24.6 | 164 | 20.2 | |
| Contact with COVID | | | | | |
| Contact | 288 | 55.7 | 634 | 78.1 | <0.001 |
| No contact | 229 | 44.3 | 178 | 21.9 | |
| Contact with Severe COVID | | | | | |
| Contact | | | 352 | 43.4 | |
| No contact | | | 460 | 56.7 | |
| Depression | | | | | |
| Probable depression | 342 | 66.2 | 430 | 53.0 | <0.001 |
| No probable depression | 175 | 33.9 | 382 | 47.0 | |
| Social Media News | | | | | |
| Yes | 381 | 73.7 | 536 | 66.0 | 0.003 |
| No | 136 | 26.3 | 276 | 34.0 | |
| Public Funded News | | | | | |
| Yes | 144 | 27.9 | 183 | 22.5 | 0.03 |
| No | 373 | 72.2 | 629 | 77.5 | |
| Commercial TV News | | | | | |
| Yes | 344 | 66.5 | 556 | 68.5 | 0.46 |
| No | 173 | 33.5 | 256 | 31.5 | |
| New York Times | | | | | |
| Yes | 206 | 39.9 | 307 | 37.8 | 0.46 |
| No | 311 | 60.2 | 505 | 62.2 | |
| Fox News | | | | | |
| Yes | 196 | 37.9 | 287 | 35.3 | 0.34 |
| No | 321 | 62.1 | 525 | 64.7 | |
| Vaccination intention (N = 816) | | | | | |
| Positive/Vaccinated | | | 713 | 87.4 | |
| Negative | | | 103 | 12.6 | |

COVID-19, probable depression, Fox News and social media as preferred sources of information (all positively associated), and self-assessed knowledge about COVID-19, contact with Chinese individuals, and publicly funded news as preferred sources (all negatively associated).

As noted in the Introduction, COVID-19 vaccines had been available for several months in the U.S. prior to the May 2021 survey. Approximately half the U.S. adult population had

**Table 5. Logistic regression results for associations with COVID-19 stigma for Waves 1 (N = 517) and 2 (N = 812).**

| | Wave 1 (N = 517) | | Wave 2 (N = 816) | |
|---|---|---|---|---|
| Outcome | Crude ORs | Adjusted ORs | Crude ORs | Adjusted ORs |
| | OR (95% CI) | OR (95% CI) | OR (95% CI) | OR (95% CI) |
| Age (Ref: 18–24 years old) | | | | |
| 25–34 years | 1.14 (0.47, 2.74) | 0.81 (0.26, 2.55) | 0.93 (0.42, 2.08) | 0.54 (0.20,1.47) |
| 35–44 years | 0.70 (0.28, 1.71) | 0.77 (0.23, 2.56) | 0.80 (0.36, 1.79) | 0.33 (0.20,1.47) |
| 45–54 years | 0.72 (0.27, 1.95) | 0.78 (0.21, 2.89) | 0.47 (0.20, 1.12) | 0.26 (0.12,0.95) |
| 55+ years | 0.51 (0.18, 1.41) | 0.81 (0.20, 3.22) | 0.40 (0.16, 0.97) | 0.74 (0.08,0.79) |
| Male Gender (Ref: Female) | 1.85 (1.27, 2.69) | 1.99 (1.19, 3.34) | 1.22 (0.92, 1.61) | 1.45 (0.97,2.18) |
| Race/Ethnicity (Ref: White Race) | | | | |
| Non-Hispanic Black | 3.45 (1.84, 6.48) | 3.68 (1.58, 8.59) | 1.84 (1.09, 3.08) | 1.19 (0.60,2.37) |
| Hispanic | 7.15 (3.99, 12.81) | 3.95 (1.84, 8.46) | 7.82 (5.00, 12.21) | 3.26 (1.82,5.85) |
| Asian | 0.68 (0.25, 1.88) | 2.10 (0.61, 7.20) | 1.65 (0.94, 2.90) | 1.46 (0.73,2.93) |
| Education (Ref: High School or less) | | | | |
| Some College | 0.54 (0.26, 1.12) | 0.50 (0.20, 1.25) | 0.74 (0.41, 1.35) | 0.55 (0.27,1.11) |
| College Degree | 3.68 (2.04, 6.63) | 2.01 (0.95, 4.28) | 3.64 (2.21, 6.02) | 1.83 (0.98,3.40) |
| Graduate School | 3.47 (1.65, 7.31) | 1.90 (0.70, 5.14) | 4.77 (2.65, 8.60) | 2.55 (1.18,5.52) |
| Employment (Ref: Unemployed) | | | | |
| Part-Time Employment | 2.45 (1.11, 5.42) | 1.51 (0.50, 4.59) | 3.61 (1.75, 7.48) | 2.92 (1.25,6.81) |
| Full-Time Employment | 3.65 (1.92, 6.93) | 1.26 (0.53, 3.00) | 6.35 (3.45, 11.66) | 2.85 (1.42,5.71) |
| COVID-19 Related Questions | | | | |
| Some/a lot of COVID-19 worry (Ref: No worry) | 2.12 (1.22, 3.68) | 1.04 (0.49, 2.23) | 7.19 (4.70, 11.01) | 3.17 (1.89,5.32) |
| A great deal COVID-19 Knowledge (Ref: Some Knowledge) | 0.37 (0.21, 0.65) | 0.85 (0.40, 1.81) | 0.14 (0.08, 0.24) | 0.49 (0.12,0.43) |
| Contact with COVID-19 (Ref: No contact) | 1.04 (0.72, 1.50) | 0.98 (0.58, 1.66) | 0.52 (0.37, 0.73) | 0.48 (0.30,0.80) |
| Contact Chinese Individuals (Ref: No contact) | 0.24 (0.14, 0.42) | 0.56 (0.28, 1.15) | 0.23 (0.16, 0.34) | 0.36 (0.21,0.63) |
| Probable Depression (Ref: No depression) | 5.97 (4.00, 8.92) | 3.63 (2.12, 6.21) | 6.35 (4.66, 8.65) | 3.37 (2.25,5.05) |
| Positive Vaccine Intentions or Vaccinated (Ref: Negative Intentions) | | | 3.11 (1.94, 5.00) | 2.17 (1.18,3.99) |
| Having any chronic conditions | | | 2.91 (2.17, 3.90) | 1.35 (0.87,2.08) |
| COVID-19 Related News Sources | | | | |
| New York Times News Source (Ref: No) | 1.08 (0.74, 1.57) | 1.14 (0.67, 1.95) | 1.17 (0.88, 1.55) | 1.04 (0.69,1.57) |
| Fox News Source (Ref: No) | 5.13 (3.25, 8.10) | 4.82 (2.69, 8.63) | 4.88 (3.57, 6.66) | 3.56 (2.33,5.42) |
| Social Media (Ref: No) | 4.24 (2.81, 6.41) | 2.22 (1.29, 3.83) | 4.40 (3.18, 6.09) | 1.98 (1.29,3.02) |
| Public News (Ref: No) | 0.19 (0.12, 0.28) | 0.27 (0.15, 0.47) | 0.31 (0.22, 0.45) | 0.33 (0.20,0.55) |
| Commercial News (Ref: No) | 0.83 (0.56, 1.22) | 0.66 (0.36, 1.20) | 1.07 (0.79, 1.44) | 1.02 (0.67,1.55) |

received at least one inoculation by the time of the May 2021 data collection. We therefore used our "vaccination status/vaccination intentions" variable for further analyses of the May 2021 survey data. As shown in Table 5, pro-vaccine attitudes (vaccinated, definitely will get vaccinated and probably will get vaccinated) versus anti-vaccines (probably will not get vaccinated and definitely will not get vaccinate) were associated with endorsing stigmatization.

Table 6 presents the SABR items by the vaccination status/intentions variable. The SABR scale as a whole and all the individual SABR items showed significant differences across vaccination status/intentions. The direction of the differences was consistent in that being vaccinated and having positive intentions to be vaccinated were associated with a greater likelihood of endorsing an individual SABR item. The "definitely will not get vaccinated" group was distinctive in their low likelihood of endorsing any of the SABR items.

Table 7 presents the relationships of the vaccination status/intention variable with selected "COVID-19 experiences and vulnerabilities" items that were associated with endorsing/not

**Table 6. SABRs by vaccine intentions for Wave 2, N = 812.**

| | Vaccinated | | Definitely/ Probably will vaccinate | | Probably will not be vaccinated | | Definitely will not be vaccinated | | p-value |
|---|---|---|---|---|---|---|---|---|---|
| | N = 557 | | N = 152 | | N = 50 | | N = 53 | | |
| | **N** | **%** | **N** | **%** | **N** | **%** | **N** | **%** | |
| COVID Stigma (5-item composite)[1] | | | | | | | | | |
| Some stigma | 277 | 49.7 | 75 | 49.3 | 15 | 30.0 | 10 | 18.9 | <0.001 |
| No stigma | 280 | 50.3 | 77 | 50.7 | 35 | 70.0 | 43 | 81.1 | |
| Should people avoid areas in the United States heavily populated by Chinese? | | | | | | | | | |
| Agree/Strongly agree | 203 | 36.5 | 56 | 36.8 | 6 | 12.0 | 4 | 7.6 | <0.001 |
| Disagree/Strongly disagree | 354 | 63.6 | 96 | 63.2 | 44 | 88.0 | 49 | 92.5 | |
| Should all Chinese by forcibly checked for COVID-19? | | | | | | | | | |
| Agree/Strongly agree | 190 | 34.1 | 44 | 29.0 | 6 | 12.0 | 2 | 3.8 | <0.001 |
| Disagree/Strongly disagree | 367 | 65.9 | 108 | 71.1 | 44 | 88.0 | 51 | 96.2 | |
| Should Chinese not be allowed to enter the United States? | | | | | | | | | |
| Agree/Strongly agree | 192 | 34.5 | 49 | 32.2 | 7 | 14.0 | 5 | 9.4 | <0.001 |
| Disagree/Strongly disagree | 365 | 65.5 | 103 | 67.8 | 43 | 86.0 | 48 | 90.6 | |
| Should Americans with COVID-19 be required to wear identification tags? | | | | | | | | | |
| Agree/Strongly agree | 221 | 39.7 | 55 | 36.2 | 9 | 18.0 | 3 | 5.7 | <0.001 |
| Disagree/Strongly disagree | 336 | 60.3 | 97 | 63.8 | 41 | 82.0 | 50 | 94.3 | |
| Should people who knowingly spread COVID-19 be executed? | | | | | | | | | |
| Agree/Strongly agree | 205 | 36.8 | 48 | 31.6 | 6 | 12.0 | 3 | 5.7 | <0.001 |
| Disagree/Strongly disagree | 352 | 63.2 | 104 | 68.4 | 44 | 88.0 | 50 | 94.3 | |

**Table 7. Selected COVID-19 experiences and vulnerabilities by vaccine intentions, Wave 2, N = 812.**

| | Vaccinated | | Definitely/ Probably will vaccinate | | Probably will not be vaccinated | | Definitely will not be vaccinated | | p-value |
|---|---|---|---|---|---|---|---|---|---|
| | N = 557 | | N = 152 | | N = 50 | | N = 53 | | |
| | **N** | **%** | **N** | **%** | **N** | **%** | **N** | **%** | |
| COVID Knowledge | | | | | | | | | |
| Some | 79 | 14.2 | 20 | 13.2 | 5 | 10.0 | 3 | 5.7 | 0.31 |
| A great deal | 478 | 85.8 | 132 | 86.8 | 45 | 90.0 | 50 | 94.3 | |
| COVID Worry | | | | | | | | | |
| Somewhat/very worried | 449 | 80.6 | 125 | 82.2 | 30 | 60.0 | 16 | 30.2 | <0.001 |
| Not at all worried | 108 | 19.4 | 27 | 17.8 | 20 | 40.0 | 37 | 69.8 | |
| Do you have any of the underlying conditions, such as diabetes, overweight, heart disease, lung/breathing diseases, that could make COVID-19 disease more severe? | | | | | | | | | |
| Yes | 253 | 45.4 | 37 | 24.3 | 12 | 24.0 | 7 | 13.2 | <0.001 |
| No | 304 | 54.6 | 115 | 75.7 | 38 | 76.0 | 46 | 86.8 | |
| Contact with Severe COVID | | | | | | | | | |
| Contact | 275 | 49.4 | 53 | 34.9 | 15 | 30.0 | 9 | 17.0 | <0.001 |
| No contact | 282 | 50.6 | 99 | 65.1 | 35 | 70.0 | 44 | 83.0 | |
| Depression | | | | | | | | | |
| Probable depression | 305 | 54.8 | 83 | 54.6 | 22 | 44.0 | 20 | 37.7 | 0.06 |
| No probable depression | 252 | 45.2 | 69 | 45.4 | 28 | 56.0 | 33 | 62.3 | |

endorsing the SABR scale. There were significant relationships between the vaccination status/ intentions variable and COVID-19 worry, less knowledge, having underlying conditions, and contact with severe COVID-19 cases and probable depression.

## Discussion

As noted in the Introduction, the primary purposes for the comparison of the pre- versus post-vaccine development surveys were to assess 1) whether the frequency of endorsing COVID-19 related stigmatization changed, and 2) whether the factors associated with endorsing COVID-19 related stigmatization changed.

### Reduction in stigmatization

There was a substantial decrease in stigmatization, both in the crude odds ratio and in the multivariable adjusted odds ratio. This can be seen as a cause for some optimism with respect to decreasing COVID-19 related stigmatization in the US.

### Continuity in factors associated with endorsing stigmatizing attitudes

The second major finding from the comparison of the 2020 and 2021 surveys were the multiple factors associated with stigmatization in both surveys. These included full time employment, Black race, Hispanic ethnicity, worry about contracting COVID-19, probable depression, and Fox News and social media as preferred sources of information (all positively associated), and self-assessed knowledge about COVID-19, contact with Chinese individuals, and publicly funded news as preferred sources (all negatively associated).

### The Health Belief Model and the SABR scale

The Health Belief Model (HBM) posits that greater perceived threat of a disease—specifically greater perceived severity and greater perceived susceptibility—will lead to greater motivation to actions that would reduce the threat of the disease [29]. Endorsing behavioral restrictions on persons with or associated with a threatening disease would thus be a straightforward prediction from the model.

A number of the factors associated with stigmatization in Table 4 are consistent with the HBM. For example, worry about contracting COVID-19 and having underlying conditions that would increase the likelihood of severe disease were positively associated with stigmatizing. Greater perceived knowledge (less uncertainty) about COVID-19 and more experience with COVID-19 were negatively associated with stigmatizing. The changes between the two surveys were also generally consistent with the HBM. For example, worry decreased between the surveys and stigmatization also decreased.

### Vaccination behaviors and intentions

The biggest change across the two surveys was the emergence of vaccination behavior and new attitudes towards vaccination, At the time of the second survey, being vaccinated or intending to be vaccinated was positively associated with endorsing stigmatizing attitudes and behavioral restrictions. At first, this may appear counterintuitive. Public health officials might have hoped that being vaccinated would greatly allay worries about contracting COVID-19. There are, multiple possible reasons for the finding.

First, our data should not be interpreted as showing an effect that vaccination did not reduce stigmatization. We did not have data from the same individuals prior to and after vaccination so that we cannot infer causation at the individual level.

The prevalence of stigmatization among all respondents in the 2020 survey was 66.0%, and the prevalence of stigmatization among respondents in the 2021 survey was 46.4%. (See Table 3). Thus, there was a net reduction of approximately 20% in survey participants endorsing stigmatizing attitudes and behavioral restrictions between the 2020 and 2021 surveys. In the 2021 survey 557/812 (69%) of the participants were vaccinated. It is possible that the 20% reduction in the prevalence of stigmatizing all came from the 31% of 2021 participants who were not vaccinated, but it would seem more likely that there was also a net reduction in endorsing stigmatization among the 2021 participants who were vaccinated.

Second, there are several plausible reasons for why those had been vaccinated might still have been quite concerned about contracting COVID-19 at the time of the second survey. The vaccines were never presented as completely effective. At the time the second survey, the CDC was still recommending that vaccinated persons wear masks in many public settings [25]. Also, the vaccinated persons did have higher prevalence of underlying conditions, more personal experience with severe cases of COVID-19, and less self-perceived knowledge (greater uncertainty) about COVID-19, all of which could contribute to the threat value of COVID-19 and lead to endorsing stigma and behavioral restrictions even if one was vaccinated.

Third, as part of the increasing polarization over the vaccines, there were increasing news stories raising doubts about the effectiveness of the vaccines and concerns about the potential harmful side effects associated with being vaccinated, which could have increased the perceived threat of COVID-19 among those who had been vaccinated [40].

Each of these factors would be consistent with the basic premise of the Health Belief Model that greater perceived threat of COVID-19 would be likely to motivate endorsing stigmatizing attitudes and behavioral restrictions on persons who associated with COVID-19.

## News sources

There has been varying coverage of COVID-19 among different news sources in the U.S., with some sources taking an approach based on science while others have taken a more xenophobic and anti-scientific approach. Specific news sources, particularly Fox News, significantly downplayed the COVID-19 pandemic, particularly in the early stages (late 2020-early 2021) and were skeptical of much of the scientific expertise surrounding the virus and transmission of COVID-19 [41, 42].

However, Fox News also spread many xenophobic messages about COVID-19. Fox News frequently referred to COVID-19 as the "Chinese Coronavirus" which acted to reinforce the negative stereotypes of Asian Americans. For examples, Fox News carried stories that the Chinese government "intentionally" released the COVID-19 virus [43–45]

These xenophobic stories likely contributed to some of the associations seen in the analysis related to Chinese individuals (questions 3–5 in the SABR scale items). While Fox News did downplay the importance of COVID-19 as a disease, which would be associated with lower stigmatization of persons associated with the disease, Fox News also emphasized the association of COVID-19 with China, which would have increased stigmatization of Chinese persons. In our sample, the xenophobic, anti-Chinese effect appears to have been much stronger that the downplay of COVID-19 effect.

## Potential causal pathways

Many of the odds ratios in the univariable logistic regressions were attenuated in the multivariable regression models, suggesting complex associations, with many variables having both direct and indirect associations with endorsing stigmatizing attitudes and behavioral restrictions. Having longitudinal individual-level data would be useful in developing a full conceptual

model of causal pathways (with specification of confounders, mediators, and moderators) for endorsing stigmatizing attitudes and behavioral restrictions. Ideally, individual-level data would be matched with significant developments in the pandemic: initial concern and lockdowns, development of vaccines, political polarization of vaccination and mask mandates, emergence of multiple variants, and emergence of "pandemic fatigue."

## The SABR scale

The SABR scale was originally developed for studying stigmatization of SARS-1 and AIDS. These were serious diseases; the case fatality rate for SARS-1 was about 14% [46] and until the development of the "cocktail" antiretroviral treatments for AIDS, its case fatality rate was over 90% [47].

HIV infection was (and still is) lifelong. At the time we first used the scale for COVID-19, August 2020, the case fatality rate was unknown, but the numbers of COVID-19 deaths in the US were quite high, approximately 450–500 per day [48].

We are currently updating the SABR scale to increase its applicability to less serious infectious diseases, such as COVID-19, which now has a very low case fatality rate.

The data in this report provide evidence for multiple aspects of construct validity for the scale. Higher worry about contracting COVID-19 was associated with greater likelihood of endorsing stigmatizing attitudes and behavioral restrictions on persons with or associated with COVID-19 which may be considered construct validity for attitudes. The scale also had construct validity for experiences, with previous exposure to severe cases of COVID-19 and having underlying conditions associated with greater endorsement of stigmatizing attitudes and behavioral restrictions. The scale also had construct validity for behavior—as deciding that one would definitely not get vaccinated can be considered behavior—and was associated with very low SABR scale scores.

The scale also showed construct validity with stigmatization theory [14] as greater contact with persons of Chinese descent as associated with less endorsement of stigmatization and behavioral restrictions of persons of Chinese descent.

This short scale cannot be considered as a comprehensive measure of the stigmatization of an infectious disease associated with a socially devalued group, but there is substantial evidence for its construct validity.

## Future research

While many of the relationships in our data are generally consistent with the Health Belief Model, we would not propose that this model can generate a full understanding of the causal pathways leading to stigmatization of COVID-19. As noted in the introduction, COVID-19 stigmatization was added to existing anti-Asian/anti-Chinese stigmatization in the U.S. and was followed by a large increase in anti-Asian hate crimes [9]. We conceived of our scale as measuring the intersection of anti-Chinese stigmatization and anti-COVID-19 stigmatization and not to measure separate components. Future research should involve adapting the scale to changes in knowledge about COVID-19, for example it clearly has a much lower case-fatality rate than SARS-1 or AIDS, but there is also the potential for symptoms persisting over long periods of time (long COVID-19) [49].

The scale also needs to be complemented with measures of factual knowledge of COVID-19, the sources of accurate, mis- and disinformation and of group affiliations. And as noted above, individual level longitudinal data, tied to developments in the epidemiology, psychology, and politics of the pandemic, are needed.

## Limitations

Several limitations of this study should be mentioned. First, we did not have a nationally representative sample in either survey. We did, however, control for demographic characteristics in our multivariable analyses. Second, we did not have the same respondents in both surveys, so we were unable to study changes in attitudes and stigmatization at the individual level. Third, we had only a single item measuring knowledge of COVID-19 and this was based on self-assessment. We were not able to assess the factual content of the respondents' self-assessed knowledge. Having an accurate assessment of the respondents' knowledge of COVID-19 and of the vaccines might be quite important in assessing relationships between news sources and stigmatizing.

These limitations are important, but we believe that our data do show an important reduction in endorsing stigmatizing attitudes and behavioral restrictions, the general applicability of the Health Belief Model, and the importance of different media as preferred news sources. The development and politicization of the vaccines has led to an additional layer of complexity in the stigmatization of persons who have COVID-19 and groups who are associated with COVID-19.

## Conclusions

There was a very substantial reduction in the endorsement of stigmatizing attitudes and behavioral restrictions between the August 2020 and May 2021 surveys. Many of the factors associated with stigmatizing were significant in both surveys. The development and partial implementation of effective vaccines did not eliminate stigmatizing and added new complexity to the patterns of stigmatization.

Generating enough public concern about a new threat to health to lead people to take appropriate actions while minimizing stigmatization of persons having or associated with the disease is always a difficult public health communication task. Achieving this balance may be especially difficult in a context of widespread mis- and disinformation, a variety of competing news sources, political polarization, and the potential for hate crimes.

## Supporting information

**S1 Checklist. Clinical studies checklist.**
(DOCX)

**S2 Checklist. Strobe checklist for cohort observational studies.**
(DOCX)

**S1 File. IRB approval from New York University.**
(PDF)

## Acknowledgments

We would like to thank the participants who provided the survey information that was used in this report from MTURK workers.

## Author Contributions

**Conceptualization:** Don C. Des Jarlais, Virginia W. Chang, Lawrence Yang.

**Data curation:** Sarah Lieff, Margaux Grivel.

**Formal analysis:** Sarah Lieff, Margaux Grivel, Chenziheng Allen Weng.

**Funding acquisition:** Virginia W. Chang, Lawrence Yang.

**Investigation:** Don C. Des Jarlais.

**Methodology:** Don C. Des Jarlais, Jonathan P. Feelemyer.

**Project administration:** Don C. Des Jarlais.

**Supervision:** Don C. Des Jarlais.

**Visualization:** Jonathan P. Feelemyer.

**Writing – original draft:** Don C. Des Jarlais, Jonathan P. Feelemyer, Lawrence Yang.

**Writing – review & editing:** Don C. Des Jarlais, Sarah Lieff, Margaux Grivel, Gabriella Meltzer, Jasmin Choi, Chenziheng Allen Weng, Jonathan P. Feelemyer, Virginia W. Chang.

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
