## [Decision Letter · Decision Letter 0]

15 Aug 2022

PONE-D-22-15278COVID-19 Stigmatization after the Development of Effective Vaccines: Vaccination Behavior, Attitudes, and News SourcesPLOS ONE

Dear Dr. Des Jarlais,

Thank you for submitting your manuscript to PLOS ONE. After careful consideration, we feel that it has merit but does not fully meet PLOS ONE’s publication criteria as it currently stands. Therefore, we invite you to submit a revised version of the manuscript that addresses the points raised during the review process. Your manuscript has been reviewed by one peer-reviewer, and their review has been appended below.  The reviewer has included several comments regarding areas of the study that could be strengthened, and highlighted areas of the study that should be clarified further.  Could you please revise the manuscript to carefully address the concerns raised? Please note that we have only been able to secure a single reviewer to assess your manuscript. We are issuing a decision on your manuscript at this point to prevent further delays in the evaluation of your manuscript. Please be aware that the editor who handles your revised manuscript might find it necessary to invite additional reviewers to assess this work once the revised manuscript is submitted. However, we will aim to proceed on the basis of this single review if possible. 

We look forward to receiving your revised manuscript.

Kind regards,

Maria Elisabeth Johanna Zalm, Ph.D

Editorial Office

PLOS ONE

Journal Requirements:

“This work was supported by New York University and was funded through a research grant from New York University’s Anti-Racism Center (20-61518-G1336-MC1039).”          

“This work was supported by New York University and was funded through a research grant from New York University’s Anti-Racism Center (20-61518-G1336-MC1039).”

“This work was supported by New York University and was funded through a research grant from New York University’s Anti-Racism Center (20-61518-G1336-MC1039).”

Reviewers' comments:

Reviewer's Responses to Questions

**Comments to the Author**

1. Is the manuscript technically sound, and do the data support the conclusions?

Reviewer #1: Yes

2. Has the statistical analysis been performed appropriately and rigorously? 

Reviewer #1: Yes

3. Have the authors made all data underlying the findings in their manuscript fully available?

Reviewer #1: Yes

4. Is the manuscript presented in an intelligible fashion and written in standard English?

Reviewer #1: Yes

5. Review Comments to the Author

Reviewer #1: Although they briefly reference the Health Beliefs model (p. 9), the current study is atheoretical and does not provide any explanation for mechanisms contributing to why and how stigma manifests and plays itself out. There are several excellent theoretical frameworks for stigma—by Rachel Smith and also by Rebecca Meisenbach, for example. This would add to the investigative strength of the current manuscript. The measurement is minimal but it is good that they compare data obtained during two point in time approximately 10 months apart.

Their participants are largely thirty something, employed, educated White males who likely read Fox News and were active on social media. Interesting commentary on Mturk, I guess. It is also interesting that they had some Asians in their samples.

Mturk is a good data source. It would have been good to obtain larger samples in both waves given the affordability of data collection on Mturk. Their comparative time frame is also appropriate. I also like that they included attention checks esp. with an Mturk master sample which does a lot of survey work.

SABR has some extreme items potentially traumatizing/harming participants arousing their bias

e.g. ‘wearing an identification tag’ (as in Nazi Germany)

execute people who knowingly spread COVID

These measures are extremes of stigma measurement.

It is alarming to me that in Wave 2, 39.7% of people who were vaccinated agreed that people with COVID should have to wear identification tags and 36.8% of vaccinated individuals believed people who knowingly spread COVID should be executed.

The opposite was true for people who refused to be vaccinated.

What did the researchers do at the end of this study to address the potential inflammation of prejudice elicited by these SABR items and the potential harm their study could have caused?

Many IRBs require that researchers when they ask questions capable of inflaming prejudice also do something to reduce it at the conclusion of the survey.

What suggestions do they have for minimizing stigmatization?

This is where a theory might have come into play.

The discussion of anti-Asian hate crimes at the end of this study is somewhat disjointed and comes out of nowhere. They ask a question did you ever experience prejudice/stigma?

They really do not develop this. Not really clear how this fits into their study. They need to make the connection more apparent.

6. PLOS authors have the option to publish the peer review history of their article (what does this mean?). If published, this will include your full peer review and any attached files.

Reviewer #1: No

---

## [Author Response · Author response to Decision Letter 0]

7 Oct 2022

Review Response for PLOS One Paper 

Journal Requirements:

Response: We would like to thank the editor and the reviewer for their thoughtful comments on the first submission We believe that we have appropriately addressed all of the comments and that the paper has been substantially improved as a result.

Response: We have reviewed the formatting information for the manuscript above and have made changes as necessary

Author Response: We have updated the information to Methods section to note that the study was approved by the New York University IRB and that affirmative indication of informed consent was required before the survey could be started. The participant was not asked to provide their name in order to protect confidentiality. The study did not include minors.

“This work was supported by New York University and was funded through a research grant from New York University’s Anti-Racism Center (20-61518-G1336-MC1039).” 

Please state what role the funders took in the study. If the funders had no role, please state: "The funders had no role.” 

Response: We have clarified the role of the funders in the updated manuscript and have added the necessary information to the funding source information previously stated in the original submission. We have also added the necessary information on the funding source update in the updated cover letter submitted in the revised files.

“This work was supported by New York University and was funded through a research grant from New York University’s Anti-Racism Center (20-61518-G1336-MC1039).”

“This work was supported by New York University and was funded through a research grant from New York University’s Anti-Racism Center (20-61518-G1336-MC1039).”

Response: We have removed the funding information statement from the manuscript file.

Response: There are important ethical and legal restrictions to publicly sharing the data from this manuscript, and we will note this in the updated data availability statement. Our survey contains Personal Health Information (PHI) which is protected under the Health Insurance Portability and Accountability Act (HIPPA), such as vaccination status and whether the individual suffers from any of the “underlying conditions” that would be likely to make a COVID infection more severe, as well as demographic characteristics. Providing public access to the data used in our analyses could threaten loss of the confidentiality of PHI, and was not to be permitted according to our proposal submitted to the IRB. 

Access to the data can be provided through an approved Data Use Agreement between our institution (New York University) and the institution with which the user is affiliated. Persons wanting to access the data should communicate with the NYU IRB to initiate a Data Use Agreement. 

Reviewers' comments:

Reviewer's Responses to Questions

Comments to the Author

1. Is the manuscript technically sound, and do the data support the conclusions?

Reviewer #1: Yes

2. Has the statistical analysis been performed appropriately and rigorously? 

Reviewer #1: Yes

3. Have the authors made all data underlying the findings in their manuscript fully available?

Reviewer #1: Yes

4. Is the manuscript presented in an intelligible fashion and written in standard English?

Reviewer #1: Yes

5. Review Comments to the Author

Reviewer #1: Although they briefly reference the Health Beliefs model (p. 9), the current study is atheoretical and does not provide any explanation for mechanisms contributing to why and how stigma manifests and plays itself out. There are several excellent theoretical frameworks for stigma—by Rachel Smith and also by Rebecca Meisenbach, for example. This would add to the investigative strength of the current manuscript. The measurement is minimal but it is good that they compare data obtained during two point in time approximately 10 months apart.

Their participants are largely thirty something, employed, educated White males who likely read Fox News and were active on social media. Interesting commentary on Mturk, I guess. It is also interesting that they had some Asians in their samples.

Response: We thank the reviewer for those comments. Smith had been cited in the Introduction to the first submission. In the Discussion section of the revised manuscript we provide more detail on the work of Meisenbach and Smith. We did find their work to be very helpful and thank the reviewer for asking that this work be included in the revised manuscript.

Mturk is a good data source. It would have been good to obtain larger samples in both waves given the affordability of data collection on Mturk. Their comparative time frame is also appropriate. I also like that they included attention checks esp. with an Mturk master sample which does a lot of survey work.

Response: We thank the reviewer for the positive comments.

SABR has some extreme items potentially traumatizing/harming participants arousing their bias

e.g. ‘wearing an identification tag’ (as in Nazi Germany)

execute people who knowingly spread COVID

These measures are extremes of stigma measurement.

It is alarming to me that in Wave 2, 39.7% of people who were vaccinated agreed that people with COVID should have to wear identification tags and 36.8% of vaccinated individuals believed people who knowingly spread COVID should be executed.

The opposite was true for people who refused to be vaccinated.

What did the researchers do at the end of this study to address the potential inflammation of prejudice elicited by these SABR items and the potential harm their study could have caused?

Many IRBs require that researchers when they ask questions capable of inflaming prejudice also do something to reduce it at the conclusion of the survey.

Response: First, we would like to thank the reviewer for these thoughtful comments.

The SABR scale has been reviewed by multiple times by multiple IRBs, including those of the Beth Israel Medical Center and of New York University, since it was first developed in 2003. All of the reviews concluded that the use of the scale was “minimal risk,” that the risk of harm was no greater than the risks in “everyday life.” (See NIH definition: "Minimal risk" means that the probability and magnitude of harm or discomfort anticipated in the research are not greater in and of themselves than those ordinarily encountered in daily life or during the performance of routine physical or psychological examinations or tests.)

The reviewer is quite correct that the SABR scale contains several extreme items. However, there has been considerable public discussion of these items. For example, during the AIDS epidemic, several state legislatures considered making deliberately infecting other persons with HIV a capital offense. There was also public discussion of requiring persons with AIDS to wear some form of badge indicating that they were infected with HIV, and of legislation that would require persons with HIV infection to inform their sexual partners that they had HIV. None of these restrictions were considered ethical or likely to be effective, so that they were not implemented, but they clearly could have been encountered by persons following the news.

More recently, there has been discussion of forcibly confining persons with COVID to their residences, which has occurred in some countries. Forcibly confining someone to their residence may be more draconian than making them wear a badge indicating that they have COVID

Finally, if one considers the internet and cable networks to be part of things “ordinarily encountered in daily life,” our dry questionnaire items are certainly much less inflammatory that the emotional rhetoric in these “aspects of daily life.”

We do recognize that several of the SABR items are extreme, but along with our IRB, do not believe that that are more inflammatory many items in the news or on the internet. 

We are considering whether to remove the extreme items in the SABR scale or add additional material at the end of the survey to address possible adverse responses to the scale and other items in the questionnaire in our future research. However, we agree with our IRBs that the present research is “minimal risk” and that the informed consent is appropriate.

What suggestions do they have for minimizing stigmatization?

This is where a theory might have come into play.

Response: We would like to thank the reviewer for this suggestion. We have added new material, specifically linked to theories of stigmatization, the revised Discussion section, where we include theoretical concepts from Meisenbach and Smith. 

The discussion of anti-Asian hate crimes at the end of this study is somewhat disjointed and comes out of nowhere. They ask a question did you ever experience prejudice/stigma?

They really do not develop this. Not really clear how this fits into their study. They need to make the connection more apparent.

Response: We agree that the mention of continuing anti-Asian hate crimes in the Discussion is “somewhat disjointed” and “appears to come “out of nowhere.” We agree that the comment on continuing anti-Asian hate crimes was “disjointed.” We asked a question about experiencing COVID-related stigmatization, but did not ask about experiencing hate crimes. We have omitted this reference to the continuation of anti-Asian hate crimes in the revision. 

We would again like to thank the editor and the reviewers for their thoughtful and constructive comments on the first submission. We believe that we have appropriately addressed all of the comments and that the manuscript has been substantially strengthened as a result.

---

## [Decision Letter · Decision Letter 1]

22 Nov 2022

PONE-D-22-15278R1COVID-19 Stigmatization after the Development of Effective Vaccines: Vaccination Behavior, Attitudes, and News SourcesPLOS ONE

Dear Dr. Des Jarlais,

Thank you for submitting your manuscript to PLOS ONE. After careful consideration, we feel that it has merit but does not fully meet PLOS ONE’s publication criteria as it currently stands. I think the paper improved already a lot. However, one of the reviewers still has some major concerns, especially with respect to the current interpretation of the news sources. In addition, there are also quite a few minor comments that should be addressed. Therefore, we invite you to submit a revised version of the manuscript that addresses the points raised during the review process.

We look forward to receiving your revised manuscript.

Kind regards,

Pieter-Paul Verhaeghe

Academic Editor

PLOS ONE

Reviewers' comments:

Reviewer's Responses to Questions

**Comments to the Author**

1. If the authors have adequately addressed your comments raised in a previous round of review and you feel that this manuscript is now acceptable for publication, you may indicate that here to bypass the “Comments to the Author” section, enter your conflict of interest statement in the “Confidential to Editor” section, and submit your "Accept" recommendation.

Reviewer #1: All comments have been addressed

Reviewer #2: All comments have been addressed

2. Is the manuscript technically sound, and do the data support the conclusions?

Reviewer #1: Yes

Reviewer #2: Yes

3. Has the statistical analysis been performed appropriately and rigorously? 

Reviewer #1: Yes

Reviewer #2: Yes

4. Have the authors made all data underlying the findings in their manuscript fully available?

Reviewer #1: Yes

Reviewer #2: Yes

5. Is the manuscript presented in an intelligible fashion and written in standard English?

Reviewer #1: Yes

Reviewer #2: Yes

6. Review Comments to the Author

Reviewer #1: (No Response)

Reviewer #2: I like the idea and implementation of the paper, but I have a number of points that should be addressed by the authors in a revision.

In general, the paper requires a bit of a language revision. By example in the last paragraph of the intro they talk about three primary questions but only two are raised and there are some typos in the paper such as on page 22 or in the headline ‘Casual Pathways’, which I suppose should be causal.

I see that they have revised the point of anti-Asian hate crime, but by mentioning it in the abstract they create the impression that they did their own analysis on it.

Overall, I like the introduction, but I wonder if the authors can broaden the context to see if the findings would apply to other countries than the US; e.g., was anti-Asian sentiment present in other countries as well.

I also think that the authors have to be more convincing when they argue for the applicability of the SABR scale. I do think that there are major differences between characteristics of the HIV and Covid-19 disease. This applies to the subsection ‘Primary outcome’ where it is stated that there were ‘strong similarities in the patterns of stigmatization’, but what does this even mean? I also agree with the other referees that the question on executing people is phrased much too strong for the context of Covid-19 and that in general these questions could have been closer to realistic restrictions, though I concur that this is based on an ex-post view from today.

A better connection between anti-Chinese sentiments in the introduction and how this leads to their inclusion in the SABR scale would be appreciated.

In the list of variables in Table 1, the rows on ‘experience of discrimination’ are not explained in the text. The authors also seem to explain the relevance of some variables by referring to results in the analyses of the survey (such as on page 9 on the knowledge on the disease); please limit this to the results section and focus on theoretical explanations why these variables should be included. And for consistency in the introduction of the categories of variables, all should state the expectation how they affect stigmatization, for some such as the vaccination behavior this is missing.

With regard to the comparison of the sample characteristics in Table 2, I don’t understand why only for some the p-value is provided; also mention in the text that you rely on p-values.

I don’t think that the answer to the main research question (1) is well presented in Table 3. The change in the SABR scale appears in the middle of the long table and I suggest to present it separately. I also suggest to look at heterogeneities in the change to make it more informative; this could also inform the subsequent analysis on the determinants. What groups within the sample does one expect to change their sentiment the most?

On page 13 the authors talk about variables ‘most strongly and consistently associated with stigmatization’, what is the objective criteria for this statement?

The authors rightfully put the health belief model into the focus. It would be great if they could link vaccination earlier to the theory; indeed, following their reasoning I was surprised about the effect of vaccination on stigmatization.

On page 19 they talk about ‘parallel changes’ in variables, what does that mean?

I’m not convinced by the section on ‘News sources’. First, they should explain the tv stations to readers who are not familiar with the US television market. Then, their reference to papers that talk about the association of Fox news and Covid-19 is not correct; as far as I see it, none of the referenced papers address Fox news (page 20). Indeed, there is a good literature on Fox news and Covid-19, but it finds exactly opposite results to the present paper. Fox news downplayed the risks of Covid-19 (Gollwitzer et al. 2020, Ash et al. 2020 and others), which according to the health belief model of the authors should lead to less stigmatization of Covid-19 cases and not more.

And lastly, I wonder how in the contribution section the participation in another paper should matter for the contribution in the current one.

Ash, E., Galletta, S., Hangartner, D., Margalit, Y. and Pinna, M. The Effect of Fox News on Health Behavior During COVID-19 (June 27, 2020). Available at SSRN: http://dx.doi.org/10.2139/ssrn.3636762

Gollwitzer, A., Martel, C., Brady, W.J. et al. Partisan differences in physical distancing are linked to health outcomes during the COVID-19 pandemic. Nat Hum Behav 4, 1186–1197 (2020). https://doi.org/10.1038/s41562-020-00977-7

7. PLOS authors have the option to publish the peer review history of their article (what does this mean?). If published, this will include your full peer review and any attached files.

Reviewer #1: No

Reviewer #2: No

---

## [Author Response · Author response to Decision Letter 1]

19 Dec 2022

First we would like to thank the reviewers for their thorough review and comments on the draft. We have made several changes and the updated draft has been improved. Below please find specific responses to comments left by the reviewers.

In general, the paper requires a bit of a language revision. By example in the last paragraph of the intro they talk about three primary questions but only two are raised and there are some typos in the paper such as on page 22 or in the headline ‘Casual Pathways’, which I suppose should be causal.

Author Response: We have gone through the paper in the revision and have corrected the language. We thank the reviewer for these comments.

I see that they have revised the point of anti-Asian hate crime, but by mentioning it in the abstract they create the impression that they did their own analysis on it.

Author response: We have removed the reference to anti-Asian hate crimes from the abstract in order to avoid the impression that we conducted analyses of anti-Asian hate crimes. 

Overall, I like the introduction, but I wonder if the authors can broaden the context to see if the findings would apply to other countries than the US; e.g., was anti-Asian sentiment present in other countries as well.

Author response: We appreciate the positive comments about the Introduction. There has been Anti-Asian sentiment in other locations outside of the US, although the US has been one of the main locations where this has been documented. In response to the reviewer comment, we have added references and note several countries that have reported these sentiments, including locations in Australia, Europe, and East Asia (i.e., South Korea). 

I also think that the authors have to be more convincing when they argue for the applicability of the SABR scale. I do think that there are major differences between characteristics of the HIV and Covid-19 disease. This applies to the subsection ‘Primary outcome’ where it is stated that there were ‘strong similarities in the patterns of stigmatization’, but what does this even mean?

Author response: We agree that HIV/AIDS and COVID-19 are very different diseases. 

The SABR scale was originally developed to assess potential similarities in the stigmatization of HIV/AIDS and SARS-1, which are also very different diseases (in terms of routes of transmission, total number of cases, geographic areas of concentration, social groups with which the disease has been associated, case fatality rate, etc.). Despite these differences, there were strong similarities in the patterns of patterns of stigmatization for SARs-1 and HIV/AIDS. All of the individual SABR items were significantly correlated for these two diseases, and many of the same factors were associated with stigmatizing both SARS-1 and HIV/AIDS. These associated factors included education, income, race/ethnicity, worry about contracting the disease, knowledge of the disease, and mental health problems. (Des Jarlais et al, 2003) 

This information on similar patterns of stigmatization show by the SABR scale across the very different diseases of SARS-1 and HIV/AIDS has now been included in the Methods section. As noted in the Methods section, we believe that the scale measures the intersection between stigmatization of a serious infectious disease and stigmatization of an associated social group that is devalued in the larger society. 

I also agree with the other referees that the question on executing people is phrased much too strong for the context of Covid-19 and that in general these questions could have been closer to realistic restrictions, though I concur that this is based on an ex-post view from today.

Author response: The question of executing persons who intentionally spread the disease came from public calls in the 1980s and 1990s in the US for executing persons who intentionally transmitted AIDS. The question was included in our 2020 survey because we wanted to use exactly the same SABR scale used in the 2003 paper comparing SARS-1 and HIV/AIDS. We repeated the same SABR scale in the 2021 survey because we wanted to assess possible change in stigmatization after the development of the effective vaccines and changing the scale would have interfered with the comparison. 

We agree with the reviewers that this question of executing people intentionally transmitting COVID-19 is probably too harsh, and have removed it from the scale in our continuing research. However, consider a case in which person A, who is infectious with COVID-19, intentionally transmits COVID-19 to person B, who is elderly and has multiple underlying conditions. We suspect many people in the US would consider this to be a case of attempted homicide. 

A better connection between anti-Chinese sentiments in the introduction and how this leads to their inclusion in the SABR scale would be appreciated.

Author response: The anti-Chinese items in the SABR scale were in the original use of the scale for assessing stigmatization of SARS-1, which was highly concentrated in China and associated with Chinese persons. The same holds for the origin of COVID-19 (SARS-CoV-2).

In the list of variables in Table 1, the rows on ‘experience of discrimination’ are not explained in the text. 

Author response: With respect to this comment, we have removed the data on the two items on experiencing discrimination related to race/ethnicity and to COVID-19 from the current report. These items were not in the 2020 survey, and thus not relevant to the comparison of the 2020 survey to the 2021 survey, nor are they closely linked to the Health Belief Model, which is the primary theoretical focus for the present paper. They were included as exploratory items in the 2021 survey to explore possible similarities in racial/ethnic discrimination and COVID-19 discrimination. Our preliminary analyses have led us to believe that we need more data on the specifics of COVID-19 discrimination (was it in response to severe COVID-19 disease or non-serious disease, or even in the absence of actual disease), and larger numbers of Black and Asian subjects (only 6% of subjects were Black and 6% were Asian) for meaningful analyses. Additionally, we need to consider the possibility that some respondents may have perceived discrimination because of their reluctance to be vaccinated. The imposition of vaccine mandates, which were publicly discussed at the time of the survey, could have created such a perception of discrimination.

The authors also seem to explain the relevance of some variables by referring to results in the analyses of the survey (such as on page 9 on the knowledge on the disease); please limit this to the results section and focus on theoretical explanations why these variables should be included. And for consistency in the introduction of the categories of variables, all should state the expectation how they affect stigmatization, for some such as the vaccination behavior this is missing. 

Author response: Almost all of the variables in Table 2 were taken from the 2003 SARS-1 HIV/AIDS study, and then repeated in the August 2020 survey. This is now clarified in the Methods section. 

With respect to the present report, our primary research questions were 1) Did the prevalence of stigmatization change after the introduction of effective vaccines? and 2) Did the correlates of stigmatization change after the introduction of the vaccines? We must emphasize that we did not have theory-based “expectations” for the answers to these two questions. Expectations would necessarily have been derived from a theoretical analysis of the effects of the vaccines, of the effects of the increasing political polarization that accompanied the introduction of the vaccines, of the continuing controversies over the origins of the virus, of the continuing increases in the deaths from the disease, and of the effects of the increase in mis- and dis-information about the vaccines. We would not have much confidence in developing “expectations” from this complex situation.

We have now clarified in the Methods section (see page 10) that we did not have expectations or hypotheses as to whether the introduction of the vaccines might lead to 1) a change in the prevalence of stigmatization or 2) changes in the correlates of stigmatization after the introduction of the vaccines. 

With regard to the comparison of the sample characteristics in Table 2, I don’t understand why only for some the p-value is provided; also mention in the text that you rely on p-values.

Author response: In Table 2, there are p-values provided for each of the variables included in the responses. For some variables, such as education, there were multiple response categories but only an overall single statistical test was used to test for a difference between the 2020 and the 2021 surveys. We have updated the manuscript to mention p-values were used to detect statistical significance in the comparisons made. 

 I don’t think that the answer to the main research question (1) is well presented in Table 3. The change in the SABR scale appears in the middle of the long table and I suggest to present it separately. I also suggest to look at heterogeneities in the change to make it more informative; this could also inform the subsequent analysis on the determinants. What groups within the sample does one expect to change their sentiment the most?

Author Response: We have reworked this section of the results and have split Table 3 into Table 3A and 3B. This allows us to separately present the data on reduction of stigmatization in Table 3A and the changes in the Health Belief Model related factors associated with stigmatization in Table 3B. We would like to thank the reviewer for calling attention to the need to present these results with greater clarity.

We are not sure what the reviewer means by “groups within the sample does one expect to change the most?” This question is worded in terms of future change, and we do not have expectations about changes in the future. If the reviewer is asking about past change (from the 2020 survey to the 2021 survey), we would first have to mention that we do not have data on the same individuals in the two surveys, so that we cannot identify individuals who did and did not change. To us, the biggest group-level change was the emergence of the vaccination/vaccination intentions groups, with 69% of the 2021 survey respondents reporting that they had received at least one immunization by May 2021. We would assume that many of those who did get vaccinated were those who in August 2020 were worried about contracting COVID-19. And we suspect that many who did get vaccinated did reduce their worry (see page 20 of the Discussion), however, getting vaccinated clearly did not eliminate being worried about contracting COVID-19 (see Table 5A)

On page 13 the authors talk about variables ‘most strongly and consistently associated with stigmatization’, what is the objective criteria for this statement?

Author Response: This wording was based on our assessment of the size of the ORs and whether a variable was associated with stigmatization in both surveys. We have changed the wording to simply note: “The variables significantly associated with stigmatization in both surveys were: full time employment, Black race, Hispanic ethnicity, worry about contracting COVID-19, probable depression, self-assessed knowledge about COVID-19, contact with Chinese individuals, and Fox News, social media and publicly-funded news as sources of information,” in Results, p.14. 

The authors rightfully put the health belief model into the focus. It would be great if they could link vaccination earlier to the theory; indeed, following their reasoning I was surprised about the effect of vaccination on stigmatization.

Author Response: We now mention the Health Belief Model in the introduction as an explanatory framework for the results. 

However, as emphasized in the Discussion, page 21 paragraph 3, we cannot assess the “effect” of vaccination at the individual level because we do not have pre-vaccination data on the prevalence of stigmatization among the individuals who did get vaccinated. 

We suspect that vaccination was probably associated with a reduction in the prevalence of stigmatization among those who were vaccinated. As noted in Discussion, p. 21, there was a large reduction in the prevalence of stigmatization between the two surveys, from 66% endorsing at least one SABR item in 2020 to 46% endorsing at least one SABR item in 2021. 

Approximately 69% of the respondents in the 2021 survey reported being vaccinated for COVID-19; assuming that prior to the vaccine rollout, those that went on to be vaccinated had similar rates of stigmatizing as those who did not receive the vaccine in 2020, the large reduction in overall stigmatizing in the 2021 survey likely included reduced stigmatizing not only among the unvaccinated, but among those who did receive the COVID-19 vaccine as well.

On page 19 they talk about ‘parallel changes’ in variables, what does that mean?

Author response: We have updated the language in this section to specifically discuss the changes among the variables and have removed the term “parallel changes.” We now note the changes in HBM related variables occurred consistent with the model. For example, on p. 19, we note that “The changes between the two surveys were also generally consistent with the HBM. For example, worry decreased between the surveys and stigmatization also decreased. 

I’m not convinced by the section on ‘News sources’. First, they should explain the tv stations to readers who are not familiar with the US television market. 

Author response: We have added more information in Table 1 to denote the revenue models for the different types of TV stations. We further described how some of these news sources are perceived in the U.S. context, specifically during the COVID-19 pandemic. We provided examples of how some of these news sources played a role in spreading information during the COVID-19 pandemic. 

Then, their reference to papers that talk about the association of Fox news and Covid-19 is not correct; as far as I see it, none of the referenced papers address Fox news (page 20). Indeed, there is a good literature on Fox news and Covid-19, but it finds exactly opposite results to the present paper. Fox news downplayed the risks of Covid-19 (Gollwitzer et al. 2020, Ash et al. 2020 and others), which according to the health belief model of the authors should lead to less stigmatization of Covid-19 cases and not more.

Author Response: While we agree with the reviewer that Fox News downplayed the virus and the COVID-19 epidemic as a whole and blamed much of the worrying on “unwarranted hysteria.” We would also note that Fox News continually engaged in xenophobic language during their newscasts. 

Fox News frequently referred to COVID-19 as the “Chinese Coronavirus” that acted to reinforce the negative stereotypes of Asian Americans. For example, Fox News carried stories that the Chinese government “intentionally” released the COVID-19 virus, and/or intentionally leaked the virus from the Wuhan lab: 1) Dorman S. Chinese virologist: China's government 'intentionally' released COVID-19. Fox News. 2020., 2) Connelly Eileen AJ. Explosive study claims to prove Chinese scientists created COVID New York Post. 2021, and 3) Blanton D. Fox News Poll: Majority believes COVID-19 leaked from lab in China. Fox News. 2020. 

These xenophobic stories likely contributed to some of the associations seen in the analysis related to Chinese individuals (questions 3-5 in the SABR scale items). While Fox News did downplay the importance of COVID-19 as a disease, which would be associated with lower stigmatization of persons associated with the disease, Fox News also emphasized the association of COVID-19 with China, which would have increased stigmatization of Chinese persons as measured by the SABR scale. In our sample, the xenophobic, anti-Chinese effect appears to have been much stronger that the downplaying of COVID-19 effect. 

And lastly, I wonder how in the contribution section the participation in another paper should matter for the contribution in the current one.

Author Response: We have reworded this section so that only co-authors of the present paper are noted in the Contribution section. The present paper contains data from two surveys, so that we wanted to acknowledge the efforts of the researchers who contributed to the initial publication of the first survey results. 

Again, we would like to thank reviewer for the careful reading and constructive comments that have substantially improved the manuscript.

---

## [Decision Letter · Decision Letter 2]

1 Feb 2023

PONE-D-22-15278R2COVID-19 Stigmatization after the Development of Effective Vaccines: Vaccination Behavior, Attitudes, and News SourcesPLOS ONE

Dear Dr. Des Jarlais,

Thank you for submitting your manuscript to PLOS ONE. After careful consideration, we feel that it has merit but does not fully meet PLOS ONE’s publication criteria as it currently stands. Therefore, we invite you to submit a revised version of the manuscript that addresses the points raised during the review process.

We look forward to receiving your revised manuscript.

Kind regards,

Ismail Ayoade Odetokun, DVM, Ph.D.

Academic Editor

PLOS ONE

Journal Requirements:

Reviewers' comments:

Reviewer's Responses to Questions

**Comments to the Author**

1. If the authors have adequately addressed your comments raised in a previous round of review and you feel that this manuscript is now acceptable for publication, you may indicate that here to bypass the “Comments to the Author” section, enter your conflict of interest statement in the “Confidential to Editor” section, and submit your "Accept" recommendation.

Reviewer #1: All comments have been addressed

Reviewer #2: (No Response)

2. Is the manuscript technically sound, and do the data support the conclusions?

Reviewer #1: Yes

Reviewer #2: Yes

3. Has the statistical analysis been performed appropriately and rigorously? 

Reviewer #1: (No Response)

Reviewer #2: Yes

4. Have the authors made all data underlying the findings in their manuscript fully available?

Reviewer #1: Yes

Reviewer #2: Yes

5. Is the manuscript presented in an intelligible fashion and written in standard English?

Reviewer #1: Yes

Reviewer #2: Yes

6. Review Comments to the Author

Reviewer #1: (No Response)

Reviewer #2: The paper has improved a lot and the authors address the suggestions well. I have only a few minor points

The section on the Methods – description of Table 1 is still a little confusing. The authors start talking about the different variables in table 1, but then shift to the ‘lack of hypotheses’ before returning to variables on ‘preferred news sources’. Please follow the order of the table, and afterwards address the other issues such as hypotheses and the statistical analysis. And if I’m not mistaken, all variable categories in Table 1 are addressed, but the ‘Experience of Discrimination related to Covid-19’ is not.

Please briefly explain the ‘Health Belief Model’ when it is mentioned for the first time in the main text (under the heading ‘Other Covid-19 related questions’). The authors do this later in the discussion section, but it helps the unfamiliar reader to know about it beforehand

The authors added a good discussion on ‘Vaccination behavior and intentions’. I wonder though if the costs of vaccination to the individual (the potential side effects of vaccines on health that were reported on in the media) could be an explaining factor for the stigmatization if one assumes that infected persons are less likely to be vaccinated.

On language and formatting: please don’t underline words or sentences, and please refrain from using terms such as ‘extreme’ in ‘extremely useful’

Once the authors have addressed the above points, I’m happy to see the paper as fit for publication.

7. PLOS authors have the option to publish the peer review history of their article (what does this mean?). If published, this will include your full peer review and any attached files.

Reviewer #1: No

Reviewer #2: No

---

## [Author Response · Author response to Decision Letter 2]

8 Feb 2023

February 6, 2023

Re: Revision Assignment for PONE-D-22-15278R2, EMID:29430ed9bc6288a5

We would like to thank the reviewer for reviewing the most recent draft and the comments below. We have addressed these points in the updated draft and believe that the paper has been strengthened as a result. Following are responses to each of the individual comments from the review. Thank you.

The section on the Methods – description of Table 1 is still a little confusing. The authors start talking about the different variables in table 1, but then shift to the ‘lack of hypotheses’ before returning to variables on ‘preferred news sources’. Please follow the order of the table, and afterwards address the other issues such as hypotheses and the statistical analysis. And if I’m not mistaken, all variable categories in Table 1 are addressed, but the ‘Experience of Discrimination related to Covid-19’ is not.

Author Response: We have modified the updated draft to move the hypothesis section lower in the paper after the description of Table 1 and before the Statistical Analysis section to allow for continuity with respect to the description of the elements of Table 1. We have added a sentence describing the “Experience of Discrimination related to COVID-19 to the Methods section where the variables are described. 

Please briefly explain the ‘Health Belief Model’ when it is mentioned for the first time in the main text (under the heading ‘Other Covid-19 related questions’). The authors do this later in the discussion section, but it helps the unfamiliar reader to know about it beforehand

Author response: The first mention of the Health Belief Model is actually at the end of the Introduction, prior to the Methods. We have added a brief description of the model to the Introduction. We now have brief descriptions of the model in the Introduction, the Methods and the Discussion, and believe that readers who are not familiar with the Health Belief Model will understand the basic premise that greater threat of the disease will lead to greater motivation towards stigmatizing and imposing behavioral restrictions on persons associated with the disease. We have added the rationale for the questions on perceived stigmatization related to COVID-19 and to race/ethnicity to the Methods section. We would note here that we did not observe any relationships between the perceived stigmatization questions and stigmatizing of persons associated with COVID-19.

The authors added a good discussion on ‘Vaccination behavior and intentions’. I wonder though if the costs of vaccination to the individual (the potential side effects of vaccines on health that were reported on in the media) could be an explaining factor for the stigmatization if one assumes that infected persons are less likely to be vaccinated.

Author Response: This is an interesting hypothesis, and we have now included wording on experiencing vaccination side effects as another possible source for stigmatizing in the Discussion. We would note a variation of this hypothesis in which persons who suffered side effects from vaccination might be resentful of public health authorities who convinced them to get vaccinated and quite willing to stigmatize the Chinese for allowing the virus to spread throughout the world. Unfortunately, our survey did not obtain sufficient data on vaccination side effects to test such hypotheses.

On language and formatting: please don’t underline words or sentences, and please refrain from using terms such as ‘extreme’ in ‘extremely useful’

Author response: We have removed underlined phrasing in the updated draft and have modified language to remove extreme statements in the updated draft.

Again, we would like to thank the reviewer for their careful reading of the revision and believe that the manuscript has been improved as a result of these comments.

---

## [Decision Letter · Decision Letter 3]

9 Mar 2023

COVID-19 Stigmatization after the Development of Effective Vaccines: Vaccination Behavior, Attitudes, and News Sources

PONE-D-22-15278R3

Dear Dr. Des Jarlais,

We’re pleased to inform you that your manuscript has been judged scientifically suitable for publication and will be formally accepted for publication once it meets all outstanding technical requirements.

Kind regards,

Ismail Ayoade Odetokun, DVM, Ph.D.

Academic Editor

PLOS ONE

Additional Editor Comments (optional):

Reviewers' comments:

Reviewer's Responses to Questions

**Comments to the Author**

1. If the authors have adequately addressed your comments raised in a previous round of review and you feel that this manuscript is now acceptable for publication, you may indicate that here to bypass the “Comments to the Author” section, enter your conflict of interest statement in the “Confidential to Editor” section, and submit your "Accept" recommendation.

Reviewer #1: All comments have been addressed

Reviewer #2: All comments have been addressed

2. Is the manuscript technically sound, and do the data support the conclusions?

Reviewer #1: Yes

Reviewer #2: Yes

3. Has the statistical analysis been performed appropriately and rigorously? 

Reviewer #1: Yes

Reviewer #2: Yes

4. Have the authors made all data underlying the findings in their manuscript fully available?

Reviewer #1: Yes

Reviewer #2: No

5. Is the manuscript presented in an intelligible fashion and written in standard English?

Reviewer #1: Yes

Reviewer #2: Yes

6. Review Comments to the Author

Reviewer #1: Revisions are spot on. I still don't like the stigma item saying people who spread COVID should be executed.

Reviewer #2: (No Response)

7. PLOS authors have the option to publish the peer review history of their article (what does this mean?). If published, this will include your full peer review and any attached files.

Reviewer #1: No

Reviewer #2: No

---

## [Editor Report · Acceptance letter]

19 Apr 2023

PONE-D-22-15278R3 

COVID-19 Stigmatization after the Development of Effective Vaccines: Vaccination Behavior, Attitudes, and News Sources 

Dear Dr. Des Jarlais:

I'm pleased to inform you that your manuscript has been deemed suitable for publication in PLOS ONE. Congratulations! Your manuscript is now with our production department. 

Kind regards, 

on behalf of

Dr. Ismail Ayoade Odetokun 

Academic Editor

PLOS ONE